# Knowledge, Attitudes, Practices, and Perceived Usability of Respirators Among Thai Healthcare Personnel During the COVID-19 Pandemic

**DOI:** 10.3390/healthcare13101186

**Published:** 2025-05-19

**Authors:** Kampanat Wangsan, Ratana Sapbamrer, Wachiranun Sirikul, Wuttipat Kiratipaisarl, Krongporn Ongprasert, Pheerasak Assavanopakun, Vithawat Surawattanasakul, Amornphat Kitro, Jinjuta Panumasvivat, Amnart Wongcharoen

**Affiliations:** 1Department of Community Medicine, Faculty of Medicine, Chiang Mai University, Chiang Mai 50200, Thailand; kampanat.w@cmu.ac.th (K.W.); wachiranun.sir@cmu.ac.th (W.S.); wuttipat.k@cmu.ac.th (W.K.); krongporn.o@cmu.ac.th (K.O.); pheerasak.assava@cmu.ac.th (P.A.); vithawat.surawat@cmu.ac.th (V.S.); amornphat.kit@cmu.ac.th (A.K.); jinjuta.p@cmu.ac.th (J.P.); 2Environmental and Occupational Medicine Excellence Center (EnOMEC), Faculty of Medicine, Chiang Mai University, Chiang Mai 50200, Thailand; 3Health Promotion Unit, Maharaj Nakorn Chiang Mai Hospital, Faculty of Medicine, Chiang Mai University, Chiang Mai 50200, Thailand; amnart.hpc@gmail.com

**Keywords:** respirator, healthcare personnel, knowledge, attitude, practice

## Abstract

**Background**: Respirators are essential for protecting healthcare personnel (HCPs) from airborne infections, and were particularly valuable during the COVID-19 pandemic. However, knowledge gaps, attitudes, and perceived usability issues may hinder their proper use, especially in settings lacking formal respiratory protection programs. **Objective**: The aim of this study was to assess the knowledge, attitudes, practices (KAP), and perceived usability of respirators among Thai healthcare personnel at a university hospital in Northern Thailand and identify differences across job roles. **Methods**: A cross-sectional survey was conducted among HCPs at a university hospital in Northern Thailand. Participants completed a validated questionnaire covering demographic data, KAP, and perceived usability of respirators. Descriptive and inferential statistics were used to analyze group differences. **Results**: A total of 479 valid responses were analyzed from physicians (31.7%), nurses (37.6%), and other HCPs (30.7%). Only around 12% of all participants correctly identified that surgical masks are not respirators, although over 90% correctly identified the nature of N95/KN95-type filtering facepiece respirators. Nurses demonstrated higher knowledge of respirator standards and proper use. Confidence and willingness to use industrial or reprocessed sterile respirators varied significantly by role (*p* < 0.05). Only 30.5% had received fit-testing. Perceived usability concerns included discomfort, heat, and breathability, reported across all groups. **Conclusions**: Knowledge, attitudes, and practices related to respirator use varied by professional role, with notable gaps in fit-testing and perceived usability. Findings highlight the need for targeted training, consistent fit-testing protocols, and improved respirator design for comfort to ensure effective respiratory protection in healthcare settings.

## 1. Introduction

The COVID-19 pandemic has led to a surge in the demand and use of personal protective equipment (PPE) worldwide, especially respirators [1,2]. Respirators are crucial in preventing HCPs from contracting airborne and droplet-transmitted infections in patient care [3,4]. Respirators (e.g., N95) are tight-fitting personal PPE designed to filter airborne particles, while medical or surgical masks are loose-fitting and intended to block large-particle droplets. To ensure proper and effective use, the Occupational Safety and Health Administration (OSHA) established a Respiratory Protection Program (RPP), outlining guidance on respirator selection, fit-testing, maintenance, and training [5]. Despite these guidelines, implementation of RPPs in the healthcare sector remains inconsistent, particularly in developing countries such as Thailand. In many institutions, policies and compliance mechanisms may be lacking. Some healthcare facilities—especially those with limited resources—have not adopted a formal RPP, increasing the risk of occupational transmission during pandemics or outbreaks. The pandemic further intensified the need for respirators and contributed to a global supply shortage. This scarcity led to the use of alternative protective strategies, including reuse, extended use, and the substitution of medical respirators with industrial-grade models [6,7]. While necessary during emergencies, these adaptations created challenges such as incorrect classification of respirators, inappropriate reuse without proper decontamination, and misuse of valved respirators, which can potentially spread unfiltered exhalations [8,9].

Understanding respirator use in real-world settings is essential for ensuring the safety of HCP. However, proper respirator usage behavior is complex and influenced by multiple factors, including workplace policy, environmental conditions, and personal knowledge and attitudes [10]. Inadequate fit, discomfort, heat, and breathing resistance often reduce adherence, particularly during extended wear or in hot and humid climates [10,11]. To better understand these factors, Knowledge, Attitudes, and Practices (KAP) studies have been widely used to assess the behavior of HCPs. Globally, several KAP studies have revealed that knowledge alone does not guarantee proper use, and gaps often exist between what workers know and how they behave in practice [8,12,13,14].

In Thailand, the infection prevention and control training related to respirator use is typically conducted at the institutional level. During the COVID-19 pandemic, hospitals often provided in-service training sessions on PPE protocols, including respirator donning and doffing. However, these trainings were not standardized nationally, and there are no mandatory licensing or certification requirements for respirator use across healthcare professions. This may contribute to variability in knowledge levels among different groups of healthcare personnel. Moreover, published data on respirator-related KAP remains limited. This study aims to assess the knowledge, attitudes, practices, and perceived usability of respirators among Thai healthcare personnel during the COVID-19 pandemic. It also seeks to compare differences across job roles to identify specific training needs or system-level gaps. The KAP questionnaire was designed based on internationally accepted guidelines, frequently reported concerns among frontline workers during the pandemic, and expert input from occupational health specialists. The findings are expected to inform future policies and training strategies to improve respiratory protection for healthcare personnel in Thailand and similar contexts.

## 2. Materials and Methods

This cross-sectional study was conducted at a university hospital in Northern Thailand. Inclusion criteria consisted of healthcare personnel (HCPs) working across various units who were at risk of respiratory infections. Although participants were drawn from different departments, specific unit information (e.g., ICU or outpatient clinics) was not collected. The individuals who declined to participate and those with medical conditions that prevented the use of tight-fitting respirators, as self-reported at the time of consent, were not included in the study. 

The study was conducted by the ethical principles of the Declaration of Helsinki and was approved by the Research Ethics Committee, Faculty of Medicine, Chiang Mai University, Thailand (Study Code: COM-2564-08382). Written informed consent was obtained from all participants before their enrollment in the study.

The required sample size was calculated using n4Studies software version 1.4.2. Assuming a finite population of approximately 2000 healthcare personnel, a 95% confidence level (corresponding to an alpha level of 0.05), a 5% margin of error, and an assumed response proportion of 0.5 (to maximize sample size), the minimum required sample size was determined to be 323 participants. A total of 542 responses were received, and after data cleaning, 479 valid responses were included in the analysis.

The questionnaire was developed based on commonly reported concerns during the pandemic and input from occupational health specialists. It comprised five main sections: (1) basic characteristics and working conditions, (2) knowledge, (3) attitudes, (4) practices, and (5) perceived usability related to respirator use. The survey included questions on general and occupational characteristics, 17 items covering knowledge (5 items), attitudes (5 items), and practices (7 items), as well as an 8-level Likert scale assessing perceived usability factors, including heat, comfort, moisture, breathability, itching, fatigue, and mask weight. Higher scores indicated greater severity of the issue. The questionnaire was constructed using examples from well-established standards, including those from NIOSH, OSHA, and relevant literature [5,8,9,15,16,17]. Content validity was validated by occupational health experts and assessed using a 4-point relevance scale, and the scale-level content validity index (S-CVI/Ave) was 0.93, indicating excellent content validity. A pilot test was conducted with 30 healthcare personnel to assess clarity and usability, after which minor modifications were made. Internal consistency reliability was assessed using Cronbach’s alpha, yielding a coefficient of 0.75, which reflects acceptable reliability.

The questionnaire was administered in Thai and was self-administered by participants. It was distributed via the REDCap online platform through the hospital’s internal organizational management system. A convenience sampling method was employed, targeting healthcare personnel at a university hospital in Northern Thailand, including physicians, nurses, and other hospital-based HCPs.

Descriptive statistics were used to summarize participant characteristics. The Shapiro–Wilk test was applied to assess the normality of continuous variables. Non-normally distributed variables were analyzed using the Kruskal–Wallis test, while ANOVA was used for variables that met the normality assumption. Chi-square or Fisher’s exact tests were used for categorical variables. Statistical significance was set at a *p*-value < 0.05. All analyses were performed using Stata version 14 (StataCorp LP, College Station, TX, USA).

## 3. Results

### 3.1. Characteristics of Participants

There were 479 valid responses, including physicians (*n* = 152, 31.7%), nurses (*n* = 180, 37.6%), and other healthcare workers (*n* = 147, 30.7%). The sex distribution differed significantly across groups (*p* < 0.001), with nurses being predominantly female (90.6%), compared to 52.6% among physicians and 72.8% among other personnel. Although the difference in median years of work experience among the three groups was not statistically significant (*p* = 0.1105), physicians had a median of 4.2 years and nurses had 5.8 years. However, working hours per week varied significantly (*p* < 0.001), with physicians reporting the highest hours (median 60 h), followed by nurses (45 h) and other personnel (41 h). Regarding environmental conditions, most participants reported working in air-conditioned environments, especially nurses (60.0%) and other healthcare personnel (61.2%), followed by physicians (42.8%). Physicians were more likely to report working in hot and dry environments (17.8%) or hot and humid conditions (21.7%), which were significantly different across groups (*p* = 0.001) (Table 1).

### 3.2. Knowledge of Respirator Use

The overall knowledge assessment revealed that nurses generally performed best, followed by physicians and other healthcare personnel (HCPs). Only around 12% of all participants correctly identified that surgical masks are not classified as respirators. In contrast, over 90% correctly identified filtering facepiece respirators (e.g., N95, KN95, KF95).

Recognition of different respirator types varied across groups. Physicians and nurses more frequently identified elastomeric full-face respirators and PAPRs compared to other HCPs (*p* = 0.033 and *p* = 0.009, respectively). More than half of all participants indicated they knew there were both medical and industrial respirators available, with no significant difference between groups (*p* = 0.696). In terms of respirator standards, the NIOSH classification (N95/N99/N100) was the most well-known, particularly among nurses (73.3%) (*p* < 0.001), while the EN standard was the least familiar across all groups. Understanding of fit-testing procedures was notably higher among physicians and nurses (approximately 75%) compared to other HCPs (30%) (*p* < 0.001). Regarding the function of valved respirators, over half of physicians and nurses correctly stated that they only protect the wearer. However, many other HCPs held incorrect beliefs or were unsure of their function (*p* < 0.001) (Table 2).

### 3.3. Attitude of the Respirator Use

Confidence in using industrial respirators instead of medical-grade ones varied significantly across roles (*p* = 0.040). Other HCPs showed the highest level of confidence (34.4%), while approximately half of all participants reported a lack of confidence overall.

A significant difference was observed in willingness to use re-sterilized respirators during shortages. Physicians (37.0%) and nurses (35.2%) were more open to reuse compared to other HCPs (25.4%) (*p* = 0.041). Most participants indicated that re-sterilized respirators should only be reused once or twice, with no significant differences across groups (*p* = 0.619).

Nurses reported the highest confidence in proper respirator use (85.4%), significantly higher than physicians (68.5%) and other HCPs (66.4%) (*p* < 0.001). Attitudes toward respirator safety also varied: physicians expressed more concern about adverse effects, while a majority of nurses and other HCPs considered respirators safe for most users (*p* < 0.001) (Table 3).

### 3.4. Practice of Respirator Use

N95/FFRs were the most commonly used respirators among all groups. Physicians reported the highest use of N95s (97.8%), while nurses had significantly more experience with PAPRs (56.7%) (*p* = 0.001). Carbon mask use was highest among other HCPs (45.7%) (*p* = 0.025). Respirator use experience varied, with most participants reporting more than one year of usage, though this was not significantly different between groups (*p* = 0.064). The frequency of respirator use was significantly higher among nurses and other HCPs, who were more likely to report daily use (*p* = 0.001). Duration of respirator wear also differed: nurses and other HCPs commonly used respirators for less than 15 min per session, while physicians reported slightly longer durations (30–60 min) (*p* = 0.044).

Nurses demonstrated the highest rate of performing fit checks consistently (61.8%) compared to physicians (37.0%) and other HCPs (42.6%) (*p* < 0.001). However, experience with formal fit-testing was low across all groups, particularly among other HCPs, where only 9.8% had ever been fit tested (*p* = 0.001) (Table 4).

### 3.5. Perceived Usability

Discomfort, heat, and breathability received the highest average perceived usability scores, followed by moisture and fatigue. Itching and mask weight received the lowest scores among all domains. When stratifying perceived usability scores by professional groups, physicians reported higher levels of discomfort, heat, moisture, breathability, itching, fatigue, and mask weight compared to other occupations. The highest scores were observed in the domains of discomfort, heat, and breathability (Figure 1).

## 4. Discussion

### 4.1. Situation and Characteristics

The study was conducted during the COVID-19 pandemic when respirators were highly used and in shortage, raising awareness of the proper respirator use. The participants were classified into three groups by job title, including physicians, nurses, and other HCPs, to find specific characteristics of different jobs. The information from the study would help to create health promotion directly on a specific group and specific issue. The majority of our participants were female, especially nurses. Sex was discussed in several studies, which has been the factor that influenced respirator usage. Females tend to fail the fit test according to facial dimension [18], particularly among Asians [19]. Remarkably, only a few of the participants had ever undergone a fit test, so the fit-testing protocol is essential among our population to ensure adequate protection against airborne.

Although the difference in median years of work experience among the professional groups was not statistically significant, physicians reported 4.2 years compared to 5.8 years among nurses. This difference, while not conclusive, may reflect variations in exposure to institutional training, familiarity with PPE protocols, and accumulated confidence in using respirators. Such differences could influence attitudes toward usability and fit, and may warrant further investigation in future studies with stratified experience-based analysis.

Most working conditions were under air conditioning, which may help ease the use of respirators. However, around 40% of physicians reported working in hot temperatures. Wearing N95 masks in warm environments can induce thermal stress, especially in individuals who are not acclimatized to such conditions [11]. Physicians reported the highest working hours, which was concordant with the longer respirator usage. Prolonged respirator usage may increase the risk of increased expiratory resistance and decreased humid permeability; changing respirators or limiting time using respirators should be considered [20].

### 4.2. Respirator Knowledge

Only around 12% of all participants accurately classified surgical masks as not being respirators. However, the participants classified the other masks correctly at a reasonably high rate, especially the filtering facepieces (FFRs). A study in India indicated a different result: 55.6% answered correctly regarding the protection properties of N95 and surgical masks [8]. This may reflect a lack of formal training or institutional guidance related to respirator classification and usage. Previous research has also shown that awareness and understanding of respirators among HCPs can be limited, especially in settings with less structured PPE training programs [10,21]. The other interesting point that arose as a result of this study revolves around knowledge about respirator standards, which nurses tend to have more of compared to other healthcare workers. The most common standard that they knew was OSHA (N95), followed by GB from China (KN95). This finding aligns with the increasing shortage of N95 masks during the COVID-19 pandemic; the importation of these masks was often reliant on China [22]. More than 75% of physicians and nurses knew about the fit test, in contrast with the other HCPs, of which only 30% were aware of fit testing. The fit-testing protocol is a part of the respiratory protection program, which has been mandatory in the USA since 1998 [23]. In Thailand, respirator fit-testing is not uniformly mandated by national regulations. While Thailand does not have a national law requiring respirator fit-testing, certain organizations, especially within the healthcare industry, have adopted such practices to enhance worker safety [24]. Our finding that only 30.5% of participants had ever undergone fit-testing highlights a critical gap in respiratory protection practices. To ensure consistent and effective protection for healthcare personnel, there is a clear need for the development and implementation of standardized fit-testing protocols at both institutional and national levels. One more interesting point is that during the shortage of respirators during the COVID pandemic, the valve respirator was used to replace the regular FFRs even though NIOSH is not recommended for infection control, as the exhalation valve allows unfiltered breath to escape, potentially spreading the pathogen [9]. About half of physicians and nurses recognized that it could protect only the wearer, while 30% of the other HCPs comprehended this condition. Similar results were observed in an Indian study, which reported that 41.7% of HCPs acknowledged the limitations of valve respirators [8].

### 4.3. Attitude Toward Respirator Usage

This study was conducted during a pandemic that caused a medical respirator shortage; substitution with industrial FFRs and re-sterile FFRs were used according to the policy of the hospital.

Approximately half of the participants reported a lack of confidence in using an industrial respirator as a substitute for a medical respirator. This finding is expected, as industrial N95 respirators provide some protection against airborne infections, but do not fully meet medical standards, particularly in terms of fluid resistance and biocompatibility. Consequently, medical N95 respirators remain the preferred choice in healthcare settings. In cases where medical-grade respirators are unavailable, such as during a pandemic-induced shortage, industrial N95s may be used with caution, provided that users receive proper training and are aware of their limitations [6,25].

Interestingly, physicians (37.0%) and nurses (35.2%) were more open to using reprocessed sterile respirators during supply shortages, compared to only 25.4% of other healthcare personnel. This difference may reflect greater familiarity or perceived necessity among clinical staff during high-demand periods. While evidence suggests that re-sterilized N95s can be used in emergencies when new respirators are unavailable, their reuse should be limited to three to five cycles, depending on the decontamination method. Furthermore, respirators must pass a fit test after each decontamination process to ensure continued protection. Despite these considerations, healthcare settings should prioritize the use of new N95 respirators whenever possible to maintain optimal safety standards [7,26,27].

Attitudes toward proper respirator use vary among healthcare professionals. The higher confidence observed among nurses (85.4%) in properly wearing respirators, compared to physicians (68.5%) and other healthcare professionals (66.4%), could be attributed to several interrelated factors such as training, experience, and workplace exposure. Nurses often have more frequent and direct patient interactions, especially in settings requiring respiratory protection, leading to greater familiarity and proficiency with respirator use [28]. Regular and structured training programs further enhance their competence, as studies have shown that targeted training improves both knowledge and confidence in respirator usage [29]. Additionally, nurses’ adaptability to extended respirator use, coupled with supportive organizational environments that prioritize infection control and provide necessary resources, fosters a culture where nurses feel more assured in their protective practices [13,30].

Physicians expressed the highest concern regarding respirator use, believing that even healthy individuals could experience adverse effects. This aligns with studies showing that prolonged respirator use can cause cardiovascular strain from increased breathing resistance and potential for hypoventilation [31,32]. Research also indicates increased heart rate, dehydration, and cognitive fatigue among HCPs wearing N95s for extended shifts [33]. In contrast, nurses and other healthcare professionals (HCPs) were more confident in respirator safety, maybe due to frequent use and familiarity. Routine respirator use may reduce perceived discomfort and concerns over potential health effects. Additionally, one-fourth of participants believed respirators only affect individuals with pre-existing conditions like pulmonary or cardiovascular disease, a perspective supported by OSHA recommendation [34]. Overall, physicians showed the greatest concern, while nurses and HCPs viewed respirators as generally safe. The varied perceptions highlight the need for further education on potential health effects across all user groups.

The study revealed significant variations in respirator use among physicians, nurses, and other healthcare professionals (HCPs), reflecting differences in workplace exposure, role-specific demands, and institutional protocols. While nearly all participants had experience using surgical masks and N95 respirators, the use of other respiratory protective equipment (RPE) varied considerably. Physicians were more familiar with filtering facepiece respirators (97.8%), while nurses had the highest engagement with powered air-purifying respirators (PAPRs) (56.7%). This is likely due to nurses having a higher frequency of direct patient care, especially with critically ill patients who require intubation, suctioning, nebulizer treatments, and mechanical ventilation—all of which generate infectious aerosols.

### 4.4. Practice of Respirator

Regarding respirator use frequency, both physicians (51.2%) and nurses (50.3%) reported using respirators less than once or twice per week, while other HCPs were less likely to fall into this category (36.1%). However, nurses (32.1%) and other HCPs (36.9%) were significantly more likely to use respirators daily compared to physicians (16.5%). Although the overall difference in use duration was statistically significant, a closer look at the data shows that nurses and HCPs reported a wider range of usage times. While a subset of these groups reported prolonged use (>8 h), many also used respirators for shorter periods (e.g., <30 min or 1–2 h). Physicians, in contrast, tended to cluster within moderate usage categories. This suggests that respirator use among nurses and HCPs is more variable, likely reflecting task diversity and shift structures rather than consistently prolonged exposure [35].

Despite frequent respirator use, all groups had evident gaps in adherence to fit-checking and fit-testing protocols. Nurses demonstrated the highest compliance with fit-checks (61.8%), while physicians and other HCPs reported lower adherence. This result could infer that nurses often receive comprehensive training in personal protective equipment (PPE), enhancing their understanding of proper procedures. A study also highlighted that nurses’ knowledge and awareness significantly influenced their compliance with PPE protocols [12].

However, actual fit-testing rates remained alarmingly low, with only a minority of participants, particularly among physicians and HCPs, ever having undergone fit-testing. This highlights a gap in respirator safety protocols, suggesting that many HCPs may unknowingly use respirators with improper fit, reducing their protective efficacy.

### 4.5. The Perceived Usability of Respirators

The perceived usability of respirators was an essential concern among participants, particularly regarding comfort, heat, and breathability. These issues align with previous research showing that prolonged respirator use, especially in hot or high-humidity environments, can cause thermal discomfort, increased breathing resistance, and facial pressure [11,20,31]. Physicians, who often reported longer hours and hotter working environments, may experience these usability challenges more intensely. Notably, this group also reported longer working hours and more frequent exposure to warm or hot environments, as shown in Table 1. Although no formal correlation analysis was conducted, these descriptive trends suggest that environmental and occupational factors may contribute to the perceived usability of respirators. Future studies incorporating multivariate or correlation analysis could provide deeper insights into these associations.

A limitation of this study is that we were asking for the overall usability of respirators, not a specific type of respirator. However, the differences in respirators could provide different experiences, as evidenced by the previous studies. While commonly used, filtering facepiece respirators (FFRs) such as N95s were associated with moderate discomfort, especially regarding facial tightness and breathing resistance during prolonged use. These perceptions are related to expiratory resistance and thermal discomfort [20,32]. While elastomeric respirators were perceived as offering a higher level of protection, users indicated more comfort and breathability than FFRs. Although they may be more durable, the bulkiness could have affected usability scores in heat management, mobility, and communication [36,37]. Reports related to powered air-purifying respirators (PAPRs) suggest improved breathability and heat comfort, potentially due to their active airflow systems. However, some participants noted drawbacks such as difficulty in communication, reduced mobility, and limited access or availability [14]. These differences underline the potential benefit of matching respirator types to specific job roles and working conditions and providing tailored training to support optimal respirator use.

### 4.6. A Key Recommendation for Improvement

Based on the study findings, several areas for improvement can be identified. One of the most urgent is the need for consistent fit-testing and practical training, especially among groups with low participation. Respirator selection should also take into account the specific tasks and environments of different HCPs, with greater attention to comfort and usability. Many participants showed gaps in understanding the difference between respirator types and the limitations of valve respirators, suggesting the need for clearer education and ongoing communication. In addition, clear protocols for safe reuse during shortages—including proper decontamination and fit checking—would help build confidence. Institutions should also consider user feedback when choosing respirators to ensure that what is provided is not only protective but also tolerable for long shifts. Finally, regular practice and simulation can go a long way in reinforcing correct use and building familiarity, especially in high-risk settings.

### 4.7. Strengths and Limitations

This study has several strengths. It included a large and diverse sample of healthcare professionals, enabling subgroup comparisons across physicians, nurses, and other HCPs. The integration of the KAP framework with usability assessment provided a comprehensive evaluation of respirator-related knowledge, attitudes, practices, and perceived challenges during a high-demand period in the COVID-19 pandemic.

Nonetheless, several limitations should be noted. The cross-sectional design limits causal inference and does not capture temporal changes. Data were self-reported, which may be subject to recall or social desirability bias. In addition, usability was assessed in general terms rather than by specific respirator models, limiting the ability to differentiate experiences across respirator types. The use of self-reported data to assess knowledge, attitudes, and practices may also have introduced social desirability bias, potentially leading to an overestimation of adherence to recommended respirator usage. Moreover, this study did not collect unit-level information (e.g., ICU, ER) on the participants’ work settings, which may have influenced their experience with respirator use. Finally, while the sample included multiple healthcare roles, findings may not be generalizable to non-clinical workers or other occupational settings.

Future studies should consider collecting data on specific hospital units or departments to enhance the applicability of findings across clinical environments. Longitudinal designs are recommended to assess the impact of interventions over time, incorporate objective measures of respirator use and fit, and evaluate the usability and physiological impact of specific respirator types. Expanding the study population to include non-clinical staff and other work environments may also enhance the generalizability of results.

## 5. Conclusions

This study highlights variations in knowledge, attitudes, practices, and perceived usability of respirators among Thai healthcare personnel during the COVID-19 pandemic, Differences in knowledge, attitude, and practice across professional roles suggest the need for tailored training and clearer guidance. Perceived usability issues related to comfort, heat, and breathability were common and may affect consistent respirator use, particularly in high-demand or warm environments. Addressing these concerns through improved respirator design, regular fit-testing, and supportive institutional policies may enhance overall adherence and protection in healthcare settings.

## Figures and Tables

**Figure 1 healthcare-13-01186-f001:**
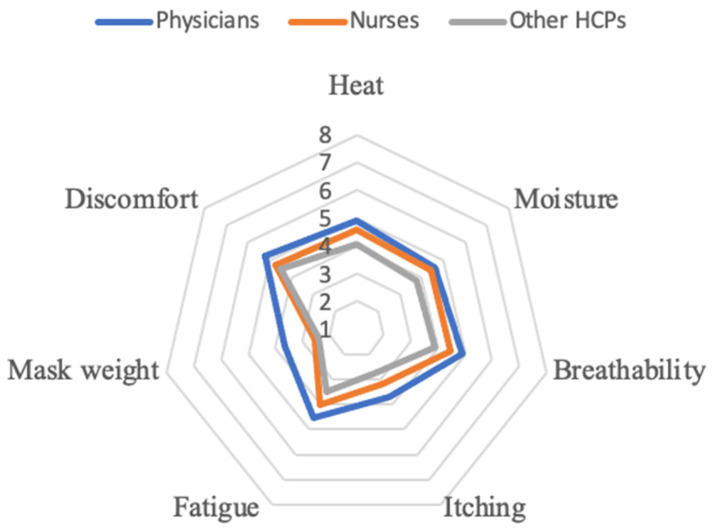
Perceived usability score of respirator among different jobs.

**Table 1 healthcare-13-01186-t001:** Characteristics of participants (*N* = 479).

Characteristics				
	Physician (*n* = 152)	Nurse (*n* = 180)	Other (*n* = 147)	*p*-Value
Sex, n (%)				
Female	80 (52.6)	163 (90.6)	107 (72.8)	<0.001 *
Male	72 (47.4)	17 (9.44)	40 (27.2)	
BMI, mean + SD.	22.6 (4.0)	22.2 (4.3)	23.9 (4.8)	0.017 *
Working experience (year), median (P25th–P75th)	4.2 (3.1–6.3)	5.8 (2.4–21.4)	6.1 (0.5–21.0)	0.1105
Working hour/week (year) median (P25th–P75th)	60 (40–72)	45 (40–56)	41 (28–56)	<0.001 *
Working condition, n (%)				
Hot and dry	27 (17.8)	27 (15.0)	9 (6.1)	0.001 *
Hot and humid	33 (21.7)	28 (15.6)	22 (15.0)	
Cold and dry	13 (8.5)	9 (5.0)	5 (3.4)	
Cold and humid	1 (0.7)	2 (1.1)	2 (1.4)	
Air-conditioning	65 (42.8)	108 (60)	90 (61.2)	
Normal temp	13 (8.5)	6 (3.3)	19 (12.9)	

* *p* < 0.05.

**Table 2 healthcare-13-01186-t002:** Knowledge of respirator use.

Questions	Answer			
	Physician (*n* = 152)	Nurse (*n* = 180)	Other (*n* = 147)	*p* Value
1. Correctly classified respirator and non-respirator				
1.1 Surgical mask	18 (12.3)	23 (13.1)	16 (12.3)	0.967
1.2 Cloth mask	88 (60.27)	89 (51.2)	77 (59.2)	0.198
1.3 Carbon mask	68 (46.9)	55 (31.4)	41 (31.8)	0.008 *
1.4 Filtering face piece (N95, Kn95, KF95)	140 (95.9)	170 (97.1)	119 (91.5)	0.083
1.5 Elastomeric half face	114 (78.1)	144 (82.3)	93 (72.1)	0.110
1.6 Elastomeric Full face	109 (74.6)	148 (84.6)	96 (73.8)	0.033 *
1.7 PAPR	124 (84.9)	160 (91.4)	103 (79.2)	0.009 *
2. Do you know that there are respirators for both industrial and medical use?				
Yes	73 (57.5)	98 (59.4)	67 (54.9)	0.696
No	22 (17.3)	24 (14.5)	26 (21.3)	
Not sure	32 (25.2)	43 (26.1)	29 (23.8)	
3. What standards do you have knowledge of?				
3.1 NIOSH (N95,N99,N100)	78 (51.3)	132 (73.3)	74 (50.3)	<0.001 *
3.2 FDA	23 (15.1)	17 (9.4)	6 (4.1)	0.005 *
3.3 ASTM	28 (18.4)	50 (20.8)	27 (18.4)	0.061
3.4 GB (KN95, KN99, KN100)	49 (32.2)	52 (29.0)	26 (17.7)	0.010 *
3.5 EN (FFP1, FFP2 FFP3)	12 (7.9)	9 (5.0)	3 (2.0)	0.064
3.6 ISO	24 (15.8)	33 (18.3)	15 (10.2)	0.109
3.7 Other	0 (0.0)	0 (0.0)	3 (2.0)	0.028 *
No	31 (20.4)	19 (10.6)	34 (23.1)	0.005 *
4. Do you know fit-testing?				
Yes	96 (75.6)	124 (75.2)	37 (30.3)	<0.001 *
Not sure	15 (11.8)	22 (13.3)	34 (27.9)	
No	16 (12.6)	19 (11.5)	51 (41.8)	
5. Can a valve respirator protect from infection?				
Protect only the user	69 (50.4)	90 (57.7)	46 (31.7)	<0.001 *
Protect both user and others	46 (33.6)	38 (24.4)	70 (48.3)	
Cannot protect both user and others	22 (16.1)	28 (17.9)	29 (20.0)	

* *p* < 0.05. Note: Item-level denominators (*n*) differ due to varying response rates. All percentages are calculated based on the number of valid responses per item.

**Table 3 healthcare-13-01186-t003:** Attitudes toward respirator use.

Question	Answer			
	Physician (*n* = 152)	Nurse (*n* = 180)	Other (*n* = 147)	*p*-Value
1. Are you confident using an industrial respirator instead of a medical one to prevent infection?				
Confidence	35 (27.6)	36 (21.8)	42 (34.4)	0.040 *
Not confidence	59 (46.5)	91 (55.1)	63 (51.6)	
Not sure/Do not know	33 (26.0)	38 (23.0)	17 (13.9)	
2. Are you willing to use a reprocessed sterile respirator during a shortage?				
Yes	47 (37.0)	58 (35.2)	31 (25.4)	0.041 *
No	46 (36.2)	75 (45.5)	67 (54.9)	
Not sure	34 (26.8)	32 (19.4)	24 (19.7)	
3. If you have to reuse a respirator, how many times you suggest reusing it?				
1	47 (37.0)	74 (44.9)	51 (41.8)	0.619
2	40 (31.5)	49 (29.7)	45 (36.9)	
3	19 (15.0)	26 (15.8)	16 (13.1)	
4	3 (2.4)	4 (2.4)	1 (0.8)	
5	12 (9.4)	7 (4.2)	5 (4.1)	
More than 5	6 (4.7)	5 (3.03)	4 (3.3)	
4. Do you think you can wear a respirator properly?				
Yes	87 (68.5)	141 (85.4)	81 (66.4)	<0.001 *
No	23 (18.1)	10 (6.1)	26 (21.3)	
Not sure	17 (13.4)	14 (8.5)	15 (12.3)	
5. Do you think wearing a respirator has any adverse health effects?				
Only for those with some health conditions	32 (25.6)	39 (23.6)	22 (18.0)	<0.001 *
Yes, for every body	61 (48.8)	56 (33.9)	35 (8.7)	
They are safe for everyone	32 (25.6)	70 (42.4)	65 (53.3)	

* *p* < 0.05. Note: Item-level denominators (*n*) differ due to varying response rates. All percentages are calculated based on the number of valid responses per item.

**Table 4 healthcare-13-01186-t004:** Practice of respirator use.

Question	Answer			
	Physician (*n* = 152)	Nurse (*n* = 180)	Other (*n* = 147)	*p*-Value
1. Which types of respiratory protective equipment have you used to prevent airborne infections?				
Surgical mask	129 (93.5)	158 (91.9)	120 (93.0)	0.895
Cloth mask	39 (28.3)	54 (31.4)	40 (31.0)	0.836
Carbon mask	51 (37.2)	52 (30.4)	59 (45.7)	0.025 *
Filtering face piece (n95)	135 (97.8)	159 (92.4)	115 (89.2)	0.012 *
Elastomeric face mask half face	32 (23.4)	24 (14.1)	25 (19.9)	0.112
Elastomeric face mask full face	26 (19.0)	23 (13.5)	22 (17.0)	0.414
PAPR	48 (35.0)	97 (56.7)	57 (44.2)	0.001 *
2. How long have you used respirators?				
Less than 1 year	46 (39.2)	59 (35.8)	46 (37.7)	0.064
1–3 years	44 (34.6)	39 (23.6)	23 (18.9)	
More than 3 years	37 (29.1)	67 (40.6)	53 (43.4)	
3. How often do you use a respirator?				
Less than 1–2/week	65 (51.2)	83 (50.3)	44 (36.1)	0.001 *
1–2/week	22 (17.3)	18 (10.9)	21 (17.2)	
3–5/week	19 (15.0)	11 (6.7)	12 (9.8)	
Every day	21 (16.5)	53 (32.1)	45 (36.9)	
4. What is the usual duration of respirator use per session?				
More than 8 h	4 (3.2)	11 (6.7)	11 (9.0)	0.044 *
4–8 h	6 (4.7)	11 (6.7)	3 (2.5)	
2–4 h	7 (5.5)	19 (11.5)	11 (9.0)	
1–2 h	30 (23.6)	28 (17.0)	24 (19.7)	
30–60 min	34 (26.8)	29 (17.6)	16 (13.1)	
15–30 min	25 (19.7)	28 (17.0)	18 (14.8)	
Less than 15 min	21 (16.6)	39 (23.6)	39 (32.0)	
5. Have you ever done a fit check before?				
Always	47 (37.0)	102 (61.8)	52 (42.6)	<0.001 *
Sometimes	50 (39.4)	47 (28.5)	37 (30.3)	
Never	16 (12.6)	12 (7.3)	21 (17.2)	
Do not know what a fit test is	14 (11.0)	4 (2.4)	12 (9.8)	
6. Have you ever had a fit test?				
Yes	36 (28.4)	36 (21.8)	12 (9.8)	0.001 *
No	91 (71.6)	129 (78.1)	110 (90.2)	
6.1 If yes, what type of fit test did you receive?				
Quantitative	9 (25)	7 (19.4)	5 (41.7)	0.093
Qualitative	8 (22.2)	3 (8.3)	1 (8.3)	
Both	2 (5.6)	3 (8.3)	3 (25.0)	
Not sure/do not know	17 (47.2)	23 (63.9)	3 (25.0)	

* *p* < 0.05. Note: Item-level denominators (n) differ due to varying response rates. All percentages are calculated based on the number of valid responses per item.

## Data Availability

Data are contained within the article.

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
