# Peer review of "Knowledge, Attitudes, Practices, and Perceived Usability of Respirators Among Thai Healthcare Personnel During the COVID-19 Pandemic"

_healthcare, 2025, doi:10.3390/healthcare13101186_

Round 1

Reviewer 1 Report

Comments and Suggestions for Authors

Dear Authors,

The study addresses a critical topic in occupational health during the COVID-19 pandemic, offering valuable insights into respirator-related KAP and usability among Thai healthcare personnel (HCPs). The cross-sectional design and subgroup comparisons by role are strengths. However, methodological clarity, data presentation consistency, and deeper contextualization of findings require improvement to meet international publication standards.

1. Abstract & Introduction

  - Objective: Specify the study’s geographical and institutional scope (e.g., "at a university hospital in Northern Thailand") to contextualize generalizability.  

  - Results: The statement "Only 12% correctly identified surgical masks are not respirators" should clarify whether this refers to all participants or specific subgroups.  

- Introduction:  

  - Gap Identification: Strengthen the rationale by explicitly citing the lack of Thai-specific KAP studies on respirators (e.g., "In Thailand, only [X] studies have examined...").  

  - Problem Statement: Link the absence of national fit-testing mandates in Thailand (mentioned later in Discussion) to the study’s urgency here.  

2. Methods  

- Sample Size Calculation:  

  - Clarity: Specify parameters used in G*Power (e.g., effect size, alpha level) to ensure reproducibility.  

- Questionnaire Validation:  

  - Detail: Describe validation steps (e.g., pilot testing, Cronbach’s alpha for reliability). Currently, "validated by occupational health experts" is vague.  

- Statistical Analysis:  

  - Clarity: Clarify why non-parametric tests (e.g., Kruskal-Wallis) were used for some variables but not others (e.g., working hours).  

3. Results  

- Tables 1–4:  

  - Consistency: Ensure denominators (e.g., n-values) are consistent. For example, Table 2 (Q1.1–1.7) lists varying denominators (e.g., 146–180), but the total sample is 479. Explain missing data or exclusions.  

  - Interpretation: In Table 1, "Working experience" has a non-significant p-value (0.1105), but the median years differ (physicians: 4.2 vs. nurses: 5.8). Discuss possible clinical relevance despite statistical non-significance.  

- Knowledge Section:  

  - Clarity: The statement "Only around 12% of participants correctly identified surgical masks..." should specify whether this refers to all HCPs or subgroups (e.g., "12% overall, with no significant differences between roles").  

4. Discussion

- Knowledge Gaps:  

  - Contextualization: The low recognition of surgical masks as non-respirators is attributed to "confusion with KN95s," but no data support this. Consider alternative explanations (e.g., lack of training) and cite relevant literature.  

- Fit Testing:  

  - Policy Implications: Expand on the absence of Thai national fit-testing mandates and how this study’s findings (e.g., 30.5% fit-testing rate) could inform policy changes.  

- Usability:  

  - Integration with Results: Link physicians’ higher discomfort scores (Figure 1) to their reported longer working hours and hotter environments (Table 1) quantitatively (e.g., correlation analysis).  

---

5. Writing & Presentation

- Clarity:  

  - Introduction: Split long sentences (e.g., "Despite these guidelines...") for readability.  

  - **Hyphenation**: Use hyphens consistently (e.g., "fit-testing" instead of "fit testing" when used adjectivally).  

- References:  

  - Relevance: Reference 21 (Heywood et al., 2021) discusses family grants, unrelated to respirators. Replace with a relevant citation.  

  - Access Dates: Add retrieval dates for online sources (e.g., FDA, WHO URLs).  

6. Limitations & Ethics

- Limitations:  

  - Discuss how self-reported data may overestimate adherence (e.g., social desirability bias).  

  - Clarify how "medical conditions preventing tight-fitting respirators" were assessed (e.g., self-reported vs. clinical evaluation).  

Comments on the Quality of English Language

Simplify complex sentences and ensure consistent terminology (e.g., "fit-testing").  

Reviewer 2 Report

Comments and Suggestions for Authors

Dear Authors,

Thank you for allowing me to review your work. Although it is an important work, I would really appreciate it if you could consider the following major and minor feedback:

1- Methods and Materials: Please consider providing validity test results of the questionnaire used to collect data in the study. Although authors stated the last three lines of page two citing studies, its important to describe test results values like  Cronbach's alpha values, pre-post test results, and the process they undertook to make changes after pre-post validity tests

Minor comment on same: Please mention the language of the questionnaire, who administered it, and how? What was the sampling process? 

2- In the methods section or subsection, please provide information on the study setting, or in results table 1, please provide information on the type of healthcare setting respondents worked in.

3- In the introduction, please provide some contextual information on infection prevention and control trainings provided during/before/ after COVID-19, specifically on respirators to the healthcare personnel. Do they exist? How do healthcare personnel acquire knowledge or skills ? E.g. Other contries have several national or local contextual trainings mandatory for nurses working in healthcare setting so existing studies attempt to gauge KAP on the basis of people who undertook the course. It would be great for the objectivity purpose, readers know some contextual background on trainings or courses or licensing, as out of KAP Knowledge is one of the components and might be different by type of healthcare personnel, as training programs bias the knowledge component values in the present study.

Much thanks!

Reviewer 3 Report

Comments and Suggestions for Authors

-I suggest including the description of respirator and mask, to health professionals who read the article differentiate them.

-In material and methods, the authors indicate the exclusion criteria, but the correct term is "elimination"; since the exclusion criteria are for longitudinal studies.

-Regarding the validation of the instrument (at the end of the last line on page 2) where it says "and was validated by occupational health experts.", the citation should be added.

-In the section on material and methods. Neither in the questionnaire nor in any part of the section on material and methods was the place where respirators are used addressed; which causes confusion because doctors report that they are in dry-warm or warm-humid places. I believe that the description of the place of use of respirators should be the same in the three groups, because they must all coincide in the place where the risk of contagion is, whether in laboratories or in the beds of hospitalized patients or in doctors' offices.

- In the discussion, page 8, line 1 reference is made to gender, but the correct thing to do is to refer to sex, in fact the authors refer to the participants as men and women.

- In the results (page 5, lines 6 and 7) it says "Physicians (37.0%) and nurses (35.2%) were more open to reuse compared to other HCPs (25.4%) (p = 0.041)." it contradicts what is stated in the discussion (first 3 lines of page 9), since it says "Among the study participants, 37% of physicians indicated a willingness to use re-sterilized N95 respirators, whereas more than half of nurses and other HCPs expressed a lack of confidence in their use." Therefore, what is discussed in that paragraph should be reconsidered

Comments on the Quality of English Language

No comments

Author Response

Please see the attatchment.

Round 2

Reviewer 3 Report

Comments and Suggestions for Authors

All the suggestions were heeded